# Polyphenols: Bioavailability, Microbiome Interactions and Cellular Effects on Health in Humans and Animals

**DOI:** 10.3390/pathogens11070770

**Published:** 2022-07-05

**Authors:** Michael B. Scott, Amy K. Styring, James S. O. McCullagh

**Affiliations:** 1Chemistry Research Laboratory, Department of Chemistry, University of Oxford, Oxford OX1 3TA, UK; michael.scott@arch.ox.ac.uk; 2School of Archaeology, University of Oxford, Oxford OX1 3TG, UK; amy.styring@arch.ox.ac.uk

**Keywords:** polyphenols, gut microbiome, pig metabolism, therapeutic effects, cytotoxicity, bone health, palaeodietary biomarkers

## Abstract

Polyphenolic compounds have a variety of functions in plants including protecting them from a range of abiotic and biotic stresses such as pathogenic infections, ionising radiation and as signalling molecules. They are common constituents of human and animal diets, undergoing extensive metabolism by gut microbiota in many cases prior to entering circulation. They are linked to a range of positive health effects, including anti-oxidant, anti-inflammatory, antibiotic and disease-specific activities but the relationships between polyphenol bio-transformation products and their interactions in vivo are less well understood. Here we review the state of knowledge in this area, specifically what happens to dietary polyphenols after ingestion and how this is linked to health effects in humans and animals; paying particular attention to farm animals and pigs. We focus on the chemical transformation of polyphenols after ingestion, through microbial transformation, conjugation, absorption, entry into circulation and uptake by cells and tissues, focusing on recent findings in relation to bone. We review what is known about how these processes affect polyphenol bioactivity, highlighting gaps in knowledge. The implications of extending the use of polyphenols to treat specific pathogenic infections and other illnesses is explored.

## 1. Introduction

Dietary polyphenols are a diverse range of organic natural products featuring multiple hydroxylated phenyl rings and are categorised into four principal classes: phenolic acids, flavonoids, stilbenes, and lignans. They are found extensively in plants where they are commonly glycosylated and perform a variety of functions including prevention of microbial infection, protection against ionising radiation and as signalling molecules with a variety of functions including growth and ripening [1,2,3,4,5]. Their prevalence across the plant kingdom makes them a common constituent of human and animal diets. Once ingested they are usually extensively metabolized by the gut microbiome in humans and animals before being absorbed into circulation, where they are treated as xenobiotics and eventually excreted. Although some polyphenols are known to adversely affect the absorption of essential nutrients (e.g., iron and other metals), they are also known for their health benefits, including anti-oxidant, anti-inflammatory and disease-specific effects such as anti-cancer activity [6,7,8,9,10,11].

Many of the reported health benefits are linked with association studies and despite a large number of in vitro studies and a smaller number of in vivo studies, the mechanisms whereby polyphenols exert their health effects are still not clearly understood. Complex biochemical and physiological processes are involved in the transition of polyphenols from dietary sources to exerting anti-oxidant, anti-inflammatory, and/or gene regulating effects in tissues. Bio-transformations are often specific to a structural class or type of polyphenol. Once absorbed from the intestine in their de-glycosylated forms, dietary polyphenols are usually (but not always) conjugated through glucuronidation, methylation or sulfation reactions, before entering circulation [12,13,14]. Subsequently, polyphenols are found in conjugated and deconjugated forms within a range of tissues in the body [15,16]. However, the relationship between these bio-transformations and therapeutic activity remains poorly understood. Furthermore, while human and lab animal (e.g., rat and mouse) responses to polyphenols are the subject of widespread experimentation, the responses in farm animals, such as pigs, poultry and cows is significantly understudied.

In this review we focus on: (1) How polyphenols are chemically modified both by the gut microbiome and, once absorbed into circulation, how this may be linked to subsequent therapeutic effects. We highlight gaps in knowledge, particularly in relation to human and animal health. (2) The health effects reported in relation to specific polyphenols with a focus on pig physiology. Here the health changes linked to polyphenolic compounds modulated by the gut microbiome are specifically highlighted, including changes in microbial species distribution, anti-microbial action against pathogens and anti-inflammatory responses. (3) Finally, in light of recent evidence showing that dietary polyphenols are incorporated into growing physiological bone in pigs [17], we consider what is known about how polyphenols are transported to different tissues and the mechanisms for their intracellular uptake. We conclude with evidence and implications of these processes for paleodietary reconstruction.

## 2. Metabolism of Dietary Polyphenols

### 2.1. Overview of Polyphenol Structure and Metabolism

Polyphenols in plants have very broad structural diversity. There are over 8000 plant-derived polyphenols recorded, however, they all share a phenolic ring and typically possess free hydroxyl groups capable of large numbers of substitutions and reactive transformations [18,19]. Examples of the different classes of polyphenols, their structures, and plant sources are presented in Table 1.

Polyphenols are treated as xenobiotics in vivo and metabolically transformed prior to excretion by animals, however, selected polyphenols can be involved in cellular metabolic processes in mammalian tissues from which specific therapeutic benefits derive [22,23]. Polyphenols start to be metabolised from the point of oral ingestion, and some polyphenols, namely flavonoids, are partially hydrolysed and in rare instances absorbed within the stomachs of monogastric mammals (Figure 1a) [14,24,25]. In general, most hydrolysis—including de-glycosylation and further biotransformation of polyphenols—occurs subsequently in the small and large intestines via mucosal and microbial enzymes, facilitating their absorption and reducing their potential toxicity (Figure 1b,c) [12,14,18,26,27,28,29,30]. Most polyphenols typically enter the colon, and undergo conjugation reactions (e.g., glucuronidation) upon uptake into intestinal epithelial cells (Figure 1d). Absorption of the aglycone form of polyphenols in the stomach or small and large intestines (approximately 5–10% of total polyphenol intake) is contingent on factors such as their hydrophobicity or lipophilicity [31,32]. Polyphenols not absorbed, including those that are unmetabolized, are excreted in feces (Figure 1e).

Nearly all polyphenols absorbed in the intestines are transported via the portal vein and subjected to glucuronidation, methylation, or sulfation in the liver as part of the natural detoxification process for xenobiotics (Figure 1f). These form conjugated polyphenol metabolites which are either then transported back to the gastrointestinal tract through the bile duct to be further metabolized (Figure 1g) and/or excreted as feces (Figure 1e), or enter circulation alongside a much lower amount of aglycone polyphenol metabolites (Figure 1h) [18,29]. Circulating polyphenols are then either excreted in urine via the kidneys, or enter other tissues (Figure 1h–i). Interestingly, polyphenols subsequently found in tissues are present in deconjugated form. This indicates a process of deconjugation in vivo, presumably via an intra-cellular process, potentially within the endoplasmic reticulum and/or may be triggered by inflammation (Section 2.3) [12]. Polyphenols consumed in high enough concentrations, beyond normal dietary levels, may enter circulation without extensive metabolism and be excreted largely in unmetabolized forms [33]. An attempt to assess the action of pomegranate polyphenols avoiding metabolism within the gut via intravenous injection found that while polyphenols did appear to accumulate in tissues such as the heart and the brain, they primarily accumulated rapidly in the kidneys following metabolism in the liver [34].

Variations in polyphenol metabolism (i.e., their biotransformation and absorption) in the gut can occur due to inter-individual and inter-species differences in gut microflora, as well as polyphenol structure [38,39]. For example, many phytoestrogenic flavonoids (i.e., genistein, apigenin, kaempferol, and naringenin) are more readily metabolised and absorbed in aglycone form, but hydrolysis and absorption of their glycosidic forms (for form most common in plants) can be more variable and dependent on the types of microflora present [40,41,42,43,44]. Furthermore, the absorbance of a small number of polyphenols in aglycone form, occurs more readily for certain types of polyphenol such as procyanidins and catechins [45,46]. In vitro research suggests that production of conjugation enzymes (e.g., uridine 5′-diphospho--glucuronlytransferase [UGT], cytochrome P450) may be induced by certain polyphenols in the colon, while in vivo research suggests that the composition of microbiota (sampled from the caecum) also influences the capacity for producing enzymes necessary for conjugation [47,48].

External factors, such as the food matrix in which polyphenols are consumed [47], as well as health status [49] can also affect the efficiency of polyphenol absorption. Age-related changes and metabolic disorders in the host have been shown to affect the distribution of intestinal microbiota, and the further ability to metabolize certain polyphenols, and the types of conjugated forms produced [49,50,51]. Host gene polymorphisms can also affect polyphenol metabolism. For example, expression of UGT and sulfotransferase enzymes required for conjugation of polyphenolic compounds in Phase II metabolism can have altered expression due to different gene polymorphisms among individuals [52,53]. Large doses of polyphenols, such as through oral administration or gavage, or from foodstuffs containing a large amount of specific compounds (e.g., epigallocatechin-gallate in green tea), have been shown to have a saturation effect on conjugation reactions, leading to greater absorption and circulation of agyclone forms rather than their conjugated counterparts [12,54,55,56].

### 2.2. The Role of the Microbiome in Polyphenol Digestion

Understanding the relationship between the particular microbiota responsible for polyphenol metabolism and the metabolised forms of polyphenols produced is key to investigating the subsequent health effects of dietary polyphenols [57]. Human studies have identified *Gordonibacter* spp. as responsible for metabolizing ellagic acid into urolithin C and intermediates (M5 and M6) [58,59]. In human and lab animal studies, *Roseburia* spp. (e.g., *R. intestinalis* XB6B4), Streptococcaceae (*Streptococcus gallolyticus* strains), *Fusobacterium* spp. (e.g., *F. nucleatum vincentii*, *F. nucleatum nucleatum*, *F. nucleatum animalis*); *Aggregatibacter* spp. (e.g., *A. actinomycetemcomitans*, *A. aphrophilus*); Coriobacteriaceae (e.g., *Slackia heliotrinireducens)*, and *Lactobacillus* spp. have all been identified as producing tannase, a key enzyme in the degradation of gallotannins [60,61,62]. *L**actobacillus* spp., *Bifidobacterium* spp., and *Clostridium* spp. have also been shown to increase in abundance in response to the ingestion of tannins, along with a reduction in proinflammatory and RNS markers (e.g., cyclooxygenase-2 [COX-2] and nitric oxide synthase) [62,63,64].

Bacteria likely to be responsible for in vivo metabolism of additional polyphenolic compounds have been isolated from the human intestine. For example, a *Clostridium* sp. produced *O*-demethylangolensin (*O*-Dma) from isoflavonoids such as daidzein [65]. *Eubacterium cellulosolvens* can hydrolyze the glycosidic bonds of some, but not all, isoflavone and flavonoid compounds [66]. Further conversion of isoflavones such as daidzein and genistein and parent compounds to equol has also been observed via dehydrogenation and reduction by *Slackia equolifaciens*, *Adlercruetzia equolifaciens*, *Eggerthella* sp., and *Asaccharobacter celatus* [67,68,69,70]. *Peptostreptococcus productus, Eggtherella lenta,* and *Atopobium* spp. have also been identified as metabolizing lignans, mainly secoisolariciresinol to entorodiol and enterolactone metabolites, which are further metabolized by *Eubacterium* spp. via catalyzation and demethylation in preparation for further dehydroxylation [71,72]. Chlorogenic acid is hydrolyzed to caffeic acid and quinic acid, and subsequently hydroxylated and reduced by gut bacteria—in the case of caffeic acid—or aromatized—in the case of quinic acid—prior to being conjugated with glycosides in the liver (producing hippuric acid) [73]. *Bifidobacterium animalis lactis, Faecalbacterium prausnitzii* and many species of *Clostridium* cluster XIVa have been identified as initiating the hydrolysis of chlorogenic acid and the subsequent breakdown and conversion in the gut (i.e., hydrogenation and dehydroxylation) of caffeic acid and quinic acid into their metabolic products, prior to conjugation with glycosides in the liver [73].

It has been argued that in general, dietary polyphenols affect the growth of probiotic gut microbiota in both pigs and other animals with implications for improving gut health. Initially it was argued that dietary polyphenols altered the gut microbiome by supplying glycosides to allow fermentation in the gut [27]. However, it has been recognised that anti-oxidant and anti-inflammatory activity may also contribute to changes in microbial species distributions, with further reduction in pathogenic species caused by polyphenols potentially promoting probiotic growth (Section 3.4) [74]. An early study investigating the potential benefits of a polyphenol supplemented diet (comprising 0.2% tea polyphenols) in pigs also identified microbiome changes from the analysis of fecal samples [75]. Thirty-day old weanlings fed a polyphenol-supplemented diet had increased levels of probiotic lactobacilli and decreased levels of pathogenic Bacteroidaceae, as well as *C. perfigens.* [75]. Bacterial species in the pig gut microbiome that are responsible for metabolizing polyphenolic compounds themselves have not been well characterized, but many additional studies have examined changes to the pig gut microbiome as a result of consuming polyphenols (Section 3.4).

Later in vivo studies examining microbiome changes in monogastric animals consistently found a reduction in pathogenic species and increased growth of probiotic species [64,74,76,77,78]. It has been shown that polyphenols in green tea (with added selenium) promote *Lactobacillus* spp. and *Bifidobacterium* spp. growth which have been shown to decrease inflammation and support metabolism in both the functional and dysfunctional gut [74]. Similarly, ellagitannins appear to promote *Lactobacillus* spp. and *Bifidobacterium* spp. growth in the human gut [78]. Some polyphenols in red wine (e.g., resveratrol) are probiotic compounds that promote *Enterococcus* spp., *Prevotella* spp., *Bacteroides* spp., *Bifodbacterium* spp., *Eggerthella* spp. and *Blautia coccoides* growth in humans, with *Bifodocaterium* spp. linked to a reduction in cholesterol triglyceride and a lower blood pressure in humans [76]. Higher levels of *Bifidobacterium* spp. and *Lactobacillus* spp. due to red wine consumption was also identified in addition to butyrate producing bacteria, leading to improvement in mucin (MUC2) expression and improved intestinal epithelial barrier function alongside a decrease in *Escherichia coli* and plasma endotoxin producers (e.g., *Enterobacter cloacae*) [79,80]. Increased mucin expression has also been linked to activation of AMP-activated protein kinase (AMPK), regulating assembly of tight junctions in the intestinal barrier and transepithelial electrical resistance, as well as an overall reduction in transcytosis of pathogenic bacteria across the gut epithelial barrier [81,82]. Cranberry extract containing large amounts of procyanidins, flavanols, and phenolic acids has been shown to promote growth of *Akkermansia* spp. in the guts of lab mice by prompting increased mucin production [64]. *Akkermansia municiphila* has been linked to formation of the lipids 2-arachidonolyglycerl and 2-oleoylglycerol, which reduce inflammation and improve gut barrier function [83]. Lastly, quercetin (Table 1) has been shown to improve gut dysbiosis in rats fed high fat diets, inhibiting the growth of bacteria linked to obesity (e.g., Erysipelotrichaceae, *Bacillus* spp., *Eubacterium cylindroides*) [77].

Polyphenols are thought to be most active in the gut (i.e., capable of activating their anti-oxidant, anti-inflammatory and anti-biotic effects; see Section 3.3) as this is where their glycosidic bonds are hydrolyzed enabling them to interact with the gut microbiome in aglycone form before undergoing conjugation reactions [36]. After being absorbed and entering circulation, polyphenolic compounds have subsequently been shown to have anti-oxidant, anti-inflammatory, and anti-biotic effects again in other organs and tissues, but this is made more complex by metabolic changes before entering circulation [14]. In the following section an overview of the processes related to transport and tissue uptake of circulating forms of polyphenols is given, along with discussion of the mechanisms that have been proposed to lead to deconjugation at the intracellular level.

### 2.3. Polyphenols in Circulation and Their Interactions with Organs, Tissues and Cellular Metabolism

Following absorption and Phase I and II metabolism, polyphenols in circulation are most commonly found as glucuronidated, methylated or sulfated conjugates, with evidence suggesting they are commonly bound non-covalently to albumin, the most abundant protein in circulation [84]. The binding is likely to occur through deprotonation of hydroxyl groups along with a variety of non-covalent interactions including electrostatic and hydrogen bonding [85,86]. The IIA domain of albumin is capable of binding to any large heterocycle with negatively charged ligands and small aromatic carboxylic acids with negatively charged and hydrophobic surfaces [87]. The bidentate ligands of catechol groups in polyphenols (e.g., quercetin) have the highest binding affinities with albumin, although this appears to be reduced in most conjugated polyphenols [85]. Polyphenols bound to albumin in circulation may still interact with other proteins/lipids via remaining unbound hydrogen atoms in this state [85]. Metabolized polyphenols are also thought to bind to other proteins, calcium, and lipids (e.g., low density lipoproteins) in circulation but these interactions have been less well studied [41]. The ability for certain polyphenols to bind with estrogen receptors, or affect phosphorylation in signal transduction pathways, for example, suggests that polyphenols may bind to the functional proteins in these pathways, potentially providing alternative transport proteins and pathways into cells [88,89].

Transport of polyphenols from the extracellular matrix to the intracellular cytosol has been shown to take place in many organs and tissues but the mechanisms involved are not well understood, although they are thought to be mediated by cell transporters [90,91]. Intracellular deconjugation of polyphenols significantly increases their anti-oxidant, anti-inflammatory, and/or other health effects in vivo compared to conjugated forms [16,23,91,92,93,94]. Flavonoids have been shown to bind in vitro to ATP binding cassette (ABC) and multi-drug resistance protein 2 (MRP2) transporters, which could also inhibit those transporters and be conducive to efflux and additional intake of flavonoids, drugs, or cytotoxins [85,90]. It has also been argued that hydrophilic conjugated polyphenols (e.g., hydroxycinnamic acids) can be substrates for organic anion transporters (OAT1 and 3), located in the kidneys, which could explain their rapid urinary excretion [95]. Polyphenolic compounds tested in vitro on cell lines, including cancerous cell lines (HT29), have been shown to still undergo metabolic changes, potentially through glucuronide transferases activated by ABC transporters [91].

In vitro studies suggest that glucuronidated, methylated, or sulfated polyphenols in circulation are deconjugated when they enter cells. The trigger for deconjugation may be related to the interplay between mitophagy and inflammatory stimuli, potentially also linked with mitochondrial dysfunction [16]. Conjugation is likely a requirement for uptake into cells before subsequent deconjugation takes place. Inflammatory responses may be capable of triggering anti-inflammatory effects mediated by dietary polyphenols in circulation via their targeted absorption and intra-cellular deconjugation in specific tissues. The enzymes required for deconjugation are largely unknown, but beta-glucuronidases are likely responsible for returning glucuronidated polyphenols to their aglycone forms [96]. The process of deconjugation of polyphenols, and their activation of anti-inflammatory or anti-oxidant effects, appears to be a directed process, as deconjugation appears to be stimulated to a greater extent in cancer cell lines and neutrophils [97]. It has recently been argued that to avoid systemic inhibition of signaling pathways necessary for cell growth, hematopoietic activity may mediate the local activation of chemopreventive polyphenolic compounds towards the tumor microenvironment [98]. In assessing the action of polyphenols (i.e., quercetin) on vascular tissues, deconjugation within cells was again argued to occur via beta-glucuronidase, likely within the cytosol, before aglycones are released from cells to once again become rapidly re-conjugated in the liver [53,99].

In vitro, quercetin glucuronides were shown to bind with cell surface proteins of macrophages by anion binding [100]. The process of deconjugation was enhanced by acidification of the extracellular environment via lactate secretion from mitochondria responding to inflammatory conditions. Interruption of the electron transport chain as a result of mitochondria dysfunction led to increased levels of intracellular Ca^2+^, which ultimately triggered beta-glucuronidase activity [16,101]. In vitro deconjugation of quercetin was then shown to inhibit inflammatory markers (COX-2) via the c-Jun N-terminal Kinase (JNK) and p38 pathway [16]. The deconjugation process for other conjugates, such as sulfates, has not been assessed, nor has sulfatase activity to the same extent as for beta-glucuronidase [100,102].

After dietary uptake, multiple diet-derived polyphenols are subsequently found in circulation, in different conjugated forms. These can be variously taken up by multiple cell types and tissues and become available in de-conjugated forms but much remains to be understood about the mechanisms involved. Arguably a better understanding of the relationship between these processes and associated health effects will help optimise both human and animal plant-based diets to promote positive health effects.

### 2.4. Digestion and Absorption of Ellagitannins, Procyanidins and Flavonoids in Pigs

Pigs are often used as a model organism for studying human biochemistry and physiological processes, and have been particularly relevant in the study of polyphenol digestion and metabolism [103,104]. Pigs are also a suitable organism for assessing the in vivo bioavailability and health effects of polyphenols in feed as an alternative to anti-biotics, which are commonly given to farm animals [36,105]. In this section we focus on the metabolism and absorption of various polyphenolic compounds in pigs, building on the general metabolic processes reported above and highlighting these processes in relation to the metabolism and absorption of ellagitannins, procyanidins and flavonoids in particular, which have variously been reported to have a range of positive health benefits [9,106,107].

Ellagitannins are hexahydroxydiphenic acid esters, likely derived from a gallotannin precursor (i.e., penta-O-galloyl-β-D-glucose), and are found in pomegranates, strawberries, raspberries, blackberries, peaches, plums, wines, and various nuts [108,109,110]. The most relevant sources of ellagitannins for pigs are walnuts and acorns, which are included in feed in some Mediterranean countries and a significant amount of research has focused on assessment of their digestibility, nutritional effects and potential positive health outcomes [104,111]. When ellagitannins are consumed by monogastric animals, hydrolysis and spontaneous lactonization of ester bonds occurs upon entering the small intestine, after earlier hydrolysis in the stomach. This results in the formation of ellagic acids, and with further bacterial scission of the alpha-pyrone ring, urolithins (Figure 1a,c) [26,104,106,108,109].

An extensive study of the modulation of ellagitannin derivatives and their metabolic fate was performed by Espín and colleagues [104], who examined ellagitannin derivatives found in various pig tissues in vivo, including at multiple points in the gastrointestinal tract. One of the major metabolic changes to ellagitannins and their derivatives in the gastrointestinal tract is the removal of hydroxyl groups via intestinal enzymes, leading to increased lipophilicity of the compounds, improving their absorption. Following the release of ellagic acids derived from ellagitannins in the jejunum, they are metabolized sequentially within the intestine into urolithins D and C, A, and finally B, being detected in the lumen in predominantly aglycone forms (Figure 1b,c) [104]. In the tissues of the small intestine, ellagic acid was not detected however, and concentrations of most urolithins were comparatively lower, while only urolithins A and B were identified in the colon. Those excreted as feces were predominantly in the form of urolithin A, as well as hydrolysable acorn tannins that did not completely metabolize. Those hydrolysable tannins were not present in urine, but conjugated forms of ellagic acid and urolithins A–D (glucoronides and methyl ethers) were. Interestingly, many of the metabolites were detected in feces and urine within 24 h after feeding on the walnut diet [104]. This is in contrast to other studies which generally observed a delay in the detection of polyphenolic metabolites in excretion products, suggested to result from the production of enzymes necessary to digest those compounds only being triggered by the presence of specific polyphenolic compounds in the colon [33,104,112,113]. This pattern of predominantly glucuronidated urolithins, and proportionally higher amounts of urolithin A and B in excretions (and more of the lower molecular weight urolithins in urine), was confirmed in subsequent human and lab animal studies [62,114,115].

The formation of glucuronide conjugates with ellagitannin metabolites (i.e., ellagic acid and urolithins A–D) first appears to occur following absorption by the epithelial barrier of the intestines and transport via the portal vein before they are further metabolized into diglucuronide or sulfate derivatives in the liver, which are either then excreted directly or enter circulation (Figure 1d,f) [104]. Biliary metabolites of urolithins were similar to those detected in urine, with the increased presence of derivatives of urolithins D and C and the addition of sulfated conjugates [104]. These derivatized metabolites likely enter enterohepatic circulation, with urolithins D and C absorbed earlier during digestion re-entering the intestine for further metabolism (i.e., lysis of hydroxyl groups) (Figure 1g). The presence of only one hydroxyl group in urolithin B allows it to form a conjugate with only one glucuronide, and allows for faster uptake and excretion (hence its excretory presence in urine only) [104]. Any ellagic acid or its derivatives present in excretions or circulation is likely the result of absorption in the stomach, given it has not been detected in small intestinal tissues. No ellagitannins were detected in other, non-intestinal, pig tissues (e.g., liver, kidney, heart, brain, lung, muscle, and subcutaneous fat), with the exception of urolithin A and B glucuronides in peripheral plasma [104]. In other animals, namely lab mice and rats, similar testing has been done on the metabolism of oral and intraperitoneal administration of ellagitannin-containing extracts, by assessing the forms present in colon and intestinal tissues, finding that conjugates of urolithins were almost exclusively present in these tissues [116,117].

Since the examination of ellagitannin metabolism [104], pig-based studies have focused on how other polyphenols are metabolized and the extent to which they occur as deconjugated forms in a greater range of tissues, despite the metabolic changes that occur to them in the gastrointestinal tract. One particular reason for this focus is that deconjugated/aglycone forms of polyphenols may be more bioactive in terms of anti-oxidant and potential health effects (Section 3), meaning the extent to which these forms occur in a variety of tissues in vivo may be indicative of how beneficial they are as a part of feed additives for example. In a 4-week feeding study, pigs were given differing doses of aglycone quercetin in their feed (25 mg/kg vs. 50 mg/kg body weight) but were found to have similar amounts of deconjugated quercetin (in addition to conjugated forms) in their tissues (comprising kidneys, liver, mid-jejunum skeletal muscle and lung tissue proximal colon, white adipose, mesentery, intestinal lymph nodes and brain) [15]. In certain tissues (i.e., colon, mesentery, diaphragm, lungs, and brain), quercetin was only found in its deconjugated form, and approximately 90% of quercetin in the liver and jejunem was deconjugated, although the mechanisms remain unclear [15]. In the same study, levels of beta-glucuronidase (which is involved in the intracellular deconjugation of polyphenol metabolites; Section 2.3) did not appear to correlate with quercetin aglycone levels in tissues, although higher levels were found in the liver, intestinal walls and kidneys (organs of primary metabolic and excretory processes) [15]. Quercetin metabolites were also identified in highest concentrations in pig liver and kidney tissues, whereas a separate study observed the highest concentrations of quercetin metabolites in brain, heart, and spleen tissues [118]. The concentration of quercetin within tissues was similar in both studies [15,118], despite the larger amount of quercetin (500 mg/kg body weight) in the latter study [118]. Since quercetin in its aglycone form was added as a supplement to regular feed in these studies, comparing how quercetin is metabolized and the forms in which it is distributed in tissues when consumed in its natural glycosidic form might further elucidate these processes.

The metabolic fate of procyanidins in grape pomace have also been assessed in pigs in a similar way to ellagitannins. Understanding the metabolism of procyanidins has been of interest due to their ability to be absorbed and to enter circulation in their original aglycone forms, unlike the predominantly conjugated forms of other polyphenolic compounds that are observed in circulation. In general, procyanidins are still intensively metabolized, often catabolized into monomers (catechin and epicatechin) by gut microflora, but some unmetabolized procyanidins are present in plasma in unconjugated forms [46]. One of the suggested reasons for the presence of unmetabolized procyanidins in plasma is their ability to bind to cell transporters, allowing for additional efflux of unmetabolized polyphenols into cells (Section 2.3) [46,90]. Conversely, binding of flavonoids (or any catechol group-containing polyphenol) with catechol-O-methyltransferase would decrease the methylation of free catechins specifically and increase their bioavailability and absorption in aglycone form [46,119]. In an earlier study assessing the absorption of different forms of procyanidins, monomers (epicatechin) were absorbed in conjugated form based on their presence in biliary excretion, However, while not as abundant, dimers were also detected exclusively as aglycones, whereas no trimers or tetramers were detected [120].

Comparing the in vivo metabolism of flavanols (such as procyanidins) with ellagitannins, and the differences in absorption in the gut (i.e., the absorption of certain aglycone procyanidins), it is clear that polyphenol structure plays an important role in polyphenol bioavailability and in what forms this occurs. Bioavailability of dietary polyphenols is linked to gut microbiome transformations [104,113], and is sensitive to alterations in the gut microbiome (including microbiota profile and microenvironment) that have been observed in pigs reared under different conditions (Section 2.2 and Section 3.4).

## 3. Health Effects of Dietary Polyphenols in Pigs and Other Animals

### 3.1. Oxidative Stress in Animals

A large amount of research has gone into the health benefits of dietary polyphenols. Much of this has focused on their anti-oxidant effects in vivo and specifically their ability to reduce reactive oxygen species (ROS) [7,121]. However, additional health benefits, often specific to certain types of polyphenol, have also been identified, ranging from anti-tumour, chemopreventative, antibiotic, anti-inflammatory and anti-ageing effects [8,10,11,30,107,122,123]. In general, the more potent bioactive effects of polyphenols are associated with the aglycone forms, and are reduced or absent in conjugated forms and when bound to albumin, as they typically are in circulation [16,85,124,125]. Thus, the polyphenol metabolites in circulation largely become bioactive as a result of the processes of uptake into cells and intracellular deconjugation (Section 2.3). In this section we will start by briefly discussing the anti-oxidant activity of polyphenols followed by reviewing additional health effects that have been identified.

Reactive oxygen species are naturally produced in mitochondria inside cells [126] and serve important cell signaling roles in relation to mitogenic responses and to defend against pathogenic diseases, often via lipid peroxidation and/or inducing cellular apoptosis through mitogen activated protein kinase pathways (MAPK) [127,128,129]. The most common form of ROS are free radicals, namely the superoxide anion (O^2−^), but other forms of ROS—particularly hydrogen peroxide (H_2_O_2_)—are also produced by reduction enzymes that regulate superoxide, such as NADPH oxidase (Nox1 and Nox2) and superoxide dismutase (SOD) [130]. Metabolic imbalances resulting from disease, infection, diet, or other stressors, can result in abnormally high levels of ROS, leading to oxidative stress and damage to cells and other biomolecules, including proteins, lipids, and DNA. Oxidative stress can lead to increases in reactive nitrogen species (RNS) such as peroxynitrite (ONOO^−^), formed by interactions between ROS and nitric oxide (NO), additionally causing lipid peroxidation [131,132]. When endogenous anti-oxidant defence systems are overwhelmed by ROS this can result in biomolecular damage that can have long-term impacts to an animal’s health.

During normal metabolic functions endogenous anti-oxidants regulate cell processes to reduce oxidative stress, acting as a defence against potential biomolecular damage from ROS [127]. Superoxide dismutase (SOD), catalase (CAT), and glutathione (GSH) are strongly associated with responses to ROS and oxidative stress in animals [133,134]. Typically SOD and/or the nonenzymatic GSH will directly scavenge free radicals to form H_2_O_2_, producing glutathione disulfide (GSSG) in the case of GSH [134,135,136]. The conversion of free radicals to H_2_O_2_ via GSH or SOD then allows for their elimination by glutathione peroxidase (GPx) or CAT, which is additionally capable of eliminating lipid peroxides [134].

Oxidative stress can also result from proinflammatory processes by causing changes in gene expression and activating certain inflammatory cytokines, such as tumour necrosis factor alpha (TNF-α), inducing increased cellular apoptosis [7,137,138,139]. Inflammatory cytokines and damage to proteins and lipids induced via ROS can cause apoptosis similar to pathways exhibited in autoimmune responses, as well as by potential mitochondrial damage and interruption of the electron transport chain [140,141].

### 3.2. The Anti-Oxidant Activity and Epigenetic Effects of Plant-Derived Polyphenols

The anti-oxidant activity of polyphenols in vivo is thought to be mainly via the scavenging of ROS, in a similar way to endogenous anti-oxidants. It has been shown that polyphenols can inhibit enzymatic activity that causes the production of ROS both in vitro and in vivo, although the mechanisms are not fully understood [121,142,143,144,145]. The chelating properties of polyphenols are also of notable importance in their ability to relieve oxidative damage brought on by circulating metals in incidences of oxidative stress. Polyphenols with catechol groups are often very good metal chelators as they are able to form bidentate ligands, and similarly, polyphenols with hydroxyl groups in peri positions on phenolic rings also make strong candidates for chelation, although pH is a key factor in their ability to bind with metals [146,147]. Fenton reactions with H_2_O_2_ and iron or copper ions can create superoxides capable of damaging biomolecules like DNA [148]. Polyphenols with catechol and/or galloyl groups that scavenge free radicals form semi-quinone compounds that are further capable of reducing Fe^3+^ ions, causing the polyphenols to become pro-oxidant quinones [148]. The ability for polyphenols to chelate may be hampered if they are conjugated with other moieties, such as sugars [146], and conversely, the anti-oxidant capabilities of polyphenols appear to be reduced in the presence of metals, as metal ligands may reduce their reactivity with free radicals [147,149]. Most of the evidence for the reduction of ROS via chelation of polyphenols with iron are based on in vitro studies, and a fairly robust body of evidence suggests that dietary polyphenols may ultimately inhibit the absorption of dietary iron [150]. Not enough evidence is currently available to indicate whether dietary polyphenols have a significant effect on the absorption of calcium in the gut, which may be of relevance to the mobilisation of calcium to extent tissues, such as bone (Section 4.1).

Beyond their capability for scavenging ROS, polyphenols can also affect gene regulation and activate transcription factors responsible for triggering endogenous anti-oxidants, such as nuclear factor erythroid 2-related factor 2 (NRF2) [151,152]. Altered gene regulation by polyphenolic compounds appears to extend to the suppression of transcription factors responsible for inflammation and tumour formation, such as activator protein 1 (AP-1) [153], as well as direct inhibition of proinflammatory cyclooxygenases and ROS lipid peroxidases [154]. Anti-oxidant and anti-inflammatory effects of polyphenols are often observed as an increase in the anti-oxidant capacity of tissues, typically in the form of increased concentrations of GPx and/or SOD, reductions in pro-inflammatory cyclooxygenases and malondialdehydes (a product of lipid peroxidation from ROS) [64].

Some polyphenols have direct chemo-preventative effects that go beyond their antioxidative effects [10,155,156,157]. For example, ellagitannins and ellagic acids (including their metabolized derivatives) have been shown to promote the activation of apoptotic pathways, such as increasing mitochondrial caspases (e.g., cyto C and Caspase 9) in human cancer cell lines and reducing their expression in non-cancer cell lines [106,141]. Ellagitannin derived metabolites may alternatively regulate the necessary cyclins (downregulating cyclins A and B1, upregulating cyclin E) for S-phase arrest to cause apoptosis in cancerous cells as well [106,141]. The anti-proliferative effects of ellagitannins towards tumour cells may be in part due to the activation of tannase-related genes [61]. Overall, a range of polyphenolic compounds appear to be capable of regulating some of the major pathways involved in tumour formation, including the p53 tumour suppression gene and through inhibition of MAPK pathways that can lead to cancer cell growth [123]. Reductions in biomarkers for inflammation and tumor formation likely extend from the epigenetic effects of polyphenols and their modulation of microRNA (miRNA) expression, DNA methylation, and histone acetylation and/or methylation [158,159,160].

Inhibition of DNA methyltransferase in human cancer cell lines when supplemented with dietary polyphenols has been shown to result in the demethylation of promoters for tumor formation or the reactivation of tumor suppression genes [161]. In these instances, polyphenols, such as epigallocatechin gallate, can inhibit DNA methyltransferase 1 (DNMT1) by binding with protein residues in a cytosine active site, preventing the entry of DNMT1 [162]. Expression of miRNAs responsible for inactivation of tumor suppression genes in cancer cell lines are also reduced in the presence of polyphenols such as resveratrol [163] and oleuropein [164]. Oleuropein, and the products it is derived from, namely olive oil, have been studied extensively for their epigenetic effects [165], and alterations in miRNA expression that coincide with increased DNA methylation are mitigated and reversed in in vivo rat models supplemented with olive oil [166]. Proliferation of human colon adenocarcinoma cells (Caco-2) supplied with separate treatments of extra virgin olive oil, an olive oil phenolic extract, and hydroxytyrosol, was reduced alongside increases in type-1 cannabinoid receptor (CB_1_) as a result of inhibition of DNA methylation at the cannabinoid receptor 1 (CNR1) promoter [167]. A similar decrease in CNR1 promoter methylation and an increase in CNR1 expression was observed in rats administered extra virgin olive oil via gavage [167].

Additional mechanisms for polyphenol immunological and anti-cancer effects appear to result from binding with cell receptors or the inhibition of histone acetylation. For example, epigallocatechin gallate can bind with cell receptors, such as zeta-chain-associated 70 kDA protein (ZAP-70) and the 67 kDA laminin receptor (67LR), the latter of which is expressed in cells involved in immune responses, such as monocytes/macrophages, mast cells, and T-cells [168,169]. Binding of epigallocatechin gallate to 67LR may inhibit human colon adenocarcinoma cell growth [170], or result in apoptosis of multiple myeloma cells [168,171]. The proposed mechanism for these anti-cancer effects is through binding of epigallocatechin gallate to 67LR, and the inhibition of myosin II regulatory light chain (MRLC) or extracellular-signal regulated kinase (ERK) 1/2 phosphorylation, as well as cytokinesis [170,172], which similarly leads to a reduction of histamine release as a response to allergy diseases [172]. Quercetin can inhibit p300 histone acetyl transferase (HAT) activity, decreasing acetylation of nuclear factor kappa B (NF-κB) and levels of inflammatory and tumor promoting enzymes (COX-2, TNF) [173,174]. Suppression of p300 HAT activity via quercetin has also been observed as decreasing acetylation of histone H3 promoter regions of the interferon gamma inducible protein 10 (IP-10) and macrophage inflammatory protein 2 (MIP-2), mitigating inflammation in intestinal epithelial cell lines [175]. In mice with induced colorectal cancer, in vivo reductions of biomarkers for tumor formation and colorectal cancer cell proliferation have also been observed following administration of polyphenols derived from foxtail millet, alongside renewal of gut microbiome diversity to that of normal mice [176].

The anti-oxidant capabilities of polyphenols have a knock-on effect of regulating micro-environments by scavenging ROS and reduce oxidative stress, resulting in reductions in inflammation, or cancer cell proliferation and the effects of other metabolic diseases (Section 4.2), while further being capable of directly interacting with proteins, enzymes, and other metabolites within immunological and metabolic pathways. The relationships between epigenetic effects of polyphenols and immune responses, anti-cancer effects, and other diseases have been reviewed extensively [158,159,169,177]). Despite this, the effects of various polyphenols in vivo are not as well tested outside of murine models, although there has been an increased focus on how supplementation of farm animals such as pigs with dietary polyphenols can result in increases in anti-oxidant activity, reductions in inflammatory biomarkers, and changes in microbiome species populations (Section 3.4).

### 3.3. Antibiotic Effects of Dietary Polyphenols

Anti-microbial activity of polyphenolic compounds has been observed in vitro targeted towards pathogenic bacterial strains [178]. Growth of *E. coli* appears to be inhibited by flavonoids, which are argued to cause topoisomerase IV-dependent DNA cleavage and inhibition to decatenation activity [179,180,181]. Polyphenols and other phytochemical compounds from *Sesbania grandifolora* have similarly been shown to inhibit the growth of multiple additional pathogenic species, including *Staphylococcus aureous*, *Shigella flexneri*, *Salmonella typhi*, and *Vibrio cholerae* [180]. The anti-microbial activity of polyphenolic compounds may also extend to regulating gene expression in pathogenic species by other means. For example, epigallocatechin gallate appeared to disrupt the c-terminal region of HFQ in *E. coli*, a pleiotropic regulator of RNA translation efficiency and decay in gram-negative bacteria [182].

It has also been proposed that the anti-microbial activity of polyphenols results from disruption of cell membranes in both gram-negative and gram-positive bacteria strains [122,183,184]. This appears to occur either through direct binding with cell membranes, or via binding with necessary functional biomolecules. For example, catechins have been shown to interact with the peptidoglycan in cell membranes of bacteria, preventing cell division [185]. Other explanations include acting as proton donors to inhibit dehydrogenase activity and disrupt proline oxidation in cell membranes [186], or binding with phospholipids and proteins, causing imbalances that create leakage in the cell membrane [187,188,189,190]. Aside from compositional changes to cell membranes, polyphenolic compounds may directly alter cell membrane charge [122], which Laporta and colleagues [188] suggest happens when polyphenols bind with phospholipid phosphate groups causing changes to the overall van der Waals interactions among phospholipid acyl changes.

Preventing biofilm formation is a means of inhibiting growth of pathogenic bacteria. Urolithins prevent biofilm formation in *Yersinia enterocoltica* by altering the synthesis of n-acylhomoserine lactones responsible for quoroum sensing processes [191]. Procyanidins have been found to have an anti-biofilm action that inhibits growth of pathogenic bacteria, such as *Porphyromonas gingivalis*, while also suppressing inflammatory cytokines produced by macrophages (interluekin-1B [IL-1B], TNF-α, IL-6 and IL-8) [192]. These effects extend to inhibiting the necessary p-fimbriated linkage of *E. coli* required for biofilm formation, as well as reduction of *E. coli*’s hydrophobicity, although the mechanism for this remains unclear [64,193,194].

The binding of polyphenols with cell transporters also exerts anti-biotic effects [195]. In particular, the ability for polyphenols to bind with and inhibit beta lactamase in drug-resistant strains of bacteria is promising for the development of strategies towards targeting such strains with anti-microbials [196]. This has led to tests to determine whether polyphenolic compounds can act together with anti-biotic drugs to improve their efficacy against resistant strains [197], although more research is needed to determine what combinations might be effective against particular strains of bacteria, and how well polyphenols and other anti-biotics work synergistically in vivo.

### 3.4. Polyphenols Can Reduce Inflammation in Pigs

High amounts of oxidative stress and inflammatory responses in farm animals can result from stress caused by environmental conditions, which create increased susceptibility to pathogens, leading to further systemic inflammation, and can cause the repression of feeding responses in animals [198]. The therapeutic effects of dietary polyphenols can be used to reduce oxidative stress and inflammation in pigs, including both those challenged by induced increases in prooxidants and those challenged with pathogenic bacterial strains [199,200]. Although these effects are targeted towards the gut, the potential for polyphenols to activate anti-microbial properties [201] and to reduce systemic inflammation beyond the gut may also factor into the health benefits of polyphenolic feeds [36]. Resistance to extreme stressors in pigs via supplementation of feed with polyphenols could improve their overall productivity, including with respect to meat production [202].

Oxidative stress can be a general symptom of gut injury for farm animals, resulting from pathogenic infection or gut dysbiosis due to metabolic changes [203], with reductions in anti-oxidant capacity and increases in lipid peroxidation often marking serious infections (e.g., sepsis) [204,205,206]. Increased production of nitric oxide also occurs during gut injury, likely due to metabolic changes in L-arginine availability that affects nitric oxide production in viscera, or higher levels of inducible nitric oxide synthase during sepsis [207,208]. It has also been suggested that high levels of nitric oxide may be a biomarker of epithelial barrier repair [209]. Oxidative stress caused by gut injury results in an immune response via hypothalamic-pituitary-adrenal axis activation of corticotrophin-releasing factor and glucocorticoids, with the latter activating mast cell receptors and creating inflammation and disruption of the intestinal barrier [36,210,211]. The autoimmune response of the gut is further regulated by T-cells maintained by gut microbiota [212]. Tied into this hypothalamic response is a reduction in appetite, that while meant to reduce feeding in animals during restoration of gut function, also reduces productivity and weight gain in farm animals [36,213,214]. Susceptibility to triggering of inflammation and autoimmune responses by pathogens and oxidative stress is typically highest in piglets post-weaning, often leading studies into the effects of polyphenols to focus on this age group [215,216,217,218]. Lactation has also been identified as eliciting an immunological response in the livers of sows, with increased activation of NF-κB regulating promotion of inflammatory cytokines, as well as leading to higher concentrations of anti-oxidants (e.g., GPx and SOD) [219,220,221]. The effects of inflammation and disruption to the gut in pigs, and piglets in particular, can be long lasting, including chronic inflammation and generally poorer health at later stages of life [36,210].

Anti-oxidant and anti-inflammatory effects of polyphenols in pigs follow the general pattern of reducing ROS and regulating endogenous anti-oxidant homeostasis or molecular responses to inflammation. Effects of grape-derived products are perhaps the most widely studied in pigs, mainly due to their abundance of flavonoids, and procyanidins in particular, which are known for their large number of functional groups and bioactive properties [222]. Silage containing 9% grape pomace has been shown to increase the concentrations of GSH in weaned piglet brain, heart, kidney, liver, lung, quadriceps, and pancreas tissues by approximately 25% or more (although not in enterocytes or plasma), with similar reductions in ROS and Thiobarbituric acid reactive substances (TBARS; a lipid peroxidation by-product) up to 50 d post-birth [223]. Tissue-specific changes in anti-oxidant enzymes in association with polyphenol supplemented diets, may be suggestive of those changes being associated with polyphenol uptake in those tissues, Higher anti-oxidant activity has also been identified in tissues such as the colon and duodenum alongside the presence of unmetabolised procyanidins in a separate study examining the effects of grape pomace in pig diet [46], suggesting tissue-specific changes in anti-oxidant enzymes may be indicative of the uptake of polyphenol metabolites by those tissues, although further investigation is required. Changes in anti-oxidant capacity were also associated with higher average daily weight gain, increases in probiotic species and reduction of pathogenic species (e.g., Enterobacteriacea and *Camplylobacter jejuni*) in feces of grape pomace fed pigs compared to the control group [223]. Improvements to daily weight gain, as well as dry matter and nitrogen digestion have also been observed in pigs fed grape pomace [224]. A separate study examined the effects of grape seed cakes (made with 5% grape seed oil extract) fed to fattening pigs, finding that they lowered the expression of inflammatory biomarkers (IL-1B, IL-6, -8, interferon-gamma [IFN-y] and TNF-α) in spleen tissue, while increasing concentrations of anti-oxidants (e.g., GPx and CAT) [225]. In sows, grape seed extract supplemented in diet was found to increase immunoglobin and hormonal activity (progesterone and estradiol levels), and overall survivability of their piglets following gestational and weaning periods [226]. Other studies have also found increases in anti-oxidant and immunological activity in serum/plasma for pigs fed grape seed extracts, but not necessarily improvements in growth performance or reduction in counts of pathogenic bacteria (e.g., *E. coli* and Clostridia) in feces [199,227].

Other polyphenol-containing food additives have been assessed for their ability to reduce oxidative stress and inflammation. For example, holly tree derived polyphenol extracts were associated with improved liver function in weaned piglets challenged with oxidative stress using diquat, as evident from heightened anti-oxidant capacity, and expression of metabolic transferases and ferroptosis mediators in the liver [228]. Resveratrol (a stilbene common to berries) added to diets reduced *Salmonella* and *E. coli* counts in the fecal content of pathogen challenged weaned piglets, additionally increasing serum immunoglobin G (IgG), reducing TNF-α levels, and appearing to improve the gut microenvironment in general due to improvements in nitrogen digestibility [229]. Similar improvements to digestibility of calcium and ether extracts have been identified in pigs fed benzoic acid and thymol additives, alongside a larger villus height:crypt depth ratio in the jejunum ileum, improved feed to gain ratio, higher concentrations of butyric acid, and higher probiotic *Lactobacillus* spp. counts [230]. Alternative feeding regimes supplemented with different forms of plant extracts or essential plant-oil derivatives have also led to reduced expression of inflammatory markers and improved digestion [200,231,232,233,234,235]. Polyphenol supplemented diets in pigs also tend to induce higher proportions of polyunsaturated fatty acids and other anti-oxidants (e.g., GSHPx, and Vitamin E) in muscle tissue, possibly by sparing those other anti-oxidants from oxidation reactions [236,237,238,239,240].

Some in vivo experiments focusing on the routing of polyphenols circumvent feeding or oral administration by means such as intraperitoneal administration [34]. Doing so controls for the interaction of polyphenols with the gut microbiome, and any associated health effects or bioavailability issues. Evidence for the presence of polyphenol metabolites associated with changes to tissues beyond the intestine has not been scrutinized to the extent of identifying endogenous biomolecular changes taking place in those tissues, despite evidence they are routed to other tissues in vivo (Section 2.3 and Section 2.4). As a result, it is not entirely clear whether changes occurring in other tissues are primarily a result of polyphenol activity in the gut, or the result of localized action related to circulating polyphenol metabolites. Studies have been limited in their association of polyphenols detected in certain tissues (alongside any potential immunological changes in those tissues) with concurrent changes in microbial populations of the gut microbiome. One area of immunological cross-talk between the gut microbiome and tissues beyond the gut is the effects of gut microbiome health on bone formation and homeostasis [241,242,243]. Recent evidence shows the presence of metabolised forms of dietary polyphenol in growing bone. This is of interest in relation to understanding bone health and from the perspective of archaeological and forensic research [17] (Section 4.3).

## 4. Polyphenols in Bone

### 4.1. Dietary Polyphenols in Growing Bone

A number of studies provide evidence that dietary polyphenols, and similar compounds, can be incorporated into physiological, growing bone [17,244,245,246,247,248]. For example, our own research has shown that polyphenolic compounds in their aglycone forms, derived from dietary sources (including ellagic acid-derived urolithins), were transported from the diet and found in pig bones [17]. The mechanism by which dietary polyphenols can accumulate in bone requires further research but it can be speculated that conjugated polyphenols in circulation may become deconjugated at sites of bone synthesis by similar mechanisms as found in other tissues [16,90]. Other compounds with similar structural properties to polyphenols bind with bone, including tetracycline and alizarin, the latter being used as a fluorochrome label for bone synthesis studies, and both are thought to bind with calcium in hydroxyapatite, the mineralised component of bone [249,250,251]. In identifying the presence of polyphenolic compounds derived from known dietary sources in pig femurs, Alldritt and colleagues [17] were unable to detect the same compounds in any surrounding soft tissues, including adipose tissue, suggesting a specific mechanism of uptake into bone, presumably from circulation. Presence of polyphenols in bone may not necessarily indicate any form of active accumulation but rather may be due to prolonged exposure to circulating forms, sufficient for transport into cells (e.g., osteoblasts or osteoclasts) followed by deconjugation and in the case of exposure to mineralising bone matrix, chelation (or similar association) with calcium as bone is being formed or resorbed [62,252]. This process would not be necessarily restricted to juveniles as bone matrix is usually in a continual process of ‘turnover’ (mineral re-adsorption followed by re-mineralisation) throughout an animal’s lifetime [253].

Studies that have observed polyphenolic compounds in soft tissues have suggested that they remain in those tissues longer than in circulation [99], and while the exact time that polyphenol metabolites remain in tissues is unclear, they have been detected in the intestine and liver of rats and mice within 1–6 h after ingestion [12]. Conversely, concentrations of polyphenol metabolites in urine peak within 2–4 h after consumption, and reach their half-life within 24 h, although this timing varies depending on the polyphenol and the individual’s physiology and health [45,254]. Due to the slow rate at which bone remodels (as low as <5% per year in cortical bone of adult humans) [253,255,256], polyphenolic compounds identified in bone may provide a long-term signal of exposure to specific dietary polyphenols.

Bone formation is primarily the result of mesenchymal stem cells found in bone marrow differentiating into osteoblasts for the generation of new bone, often acting upon, and in response to, stress and injury [257,258]. Osteoblasts form bone by secreting osteoid, a dense organic matrix composed of Type I collagen and other proteins made of acidic amino acids, including osteocalcin, osteopontin, and bone sialoprotein [259]. Osteoid subsequently calcifies, undergoing a process in which it is overlain with a serum calcium phosphate that mineralizes into hydroxyapatite to form a crystalline structure. The most likely sites for polyphenols to accumulate in bone would be either associated with Type I collagen protein that forms most of the organic portion of the bone matrix, and/or the calcium in hydroxyapatite forming the mineral portion of the matrix.

Studies identifying tetracycline and alizarin in bone have argued that these molecules are bound to calcium specifically. In vitro experiments have demonstrated that a range of polyphenolic compounds bind directly with free calcium and hydroxyapatite, and polyphenol interactions with hydroxyapatite are likely the result of chelation with calcium ions [17,147]. At physiological pH (~7.6), polyphenols are generally deprotonated and generate oxygen centres with high charge densities, making them capable of forming hard ligands with transition metals, typically via catechol groups (Section 3.2) [146]. Carboxylate groups have more recently been argued as better sites for metal ligands, at least in relation to calcium [147].

The timing of potential chelation between polyphenols and calcium is key to understanding how they might be deposited into bone if they are bound to hydroxyapatite. Osteoblasts express specific transporters for the deposition of both calcium and phosphate [260]. It is argued that these are unable to co-transport low molecular weight molecules such as polyphenols [259]. Osteoblasts (and other types of cells) do however possess special transporters, such as ABC transporters and solute carrier 21 and 22 (SLC21/SLC22) transporters that are capable of moving different types of metabolites, including drugs [261,262]. These transporters represent one potential mechanism for the movement of polyphenols that may be present within osteoblasts to the extracellular fluid that contains nucleating hydroxyapatite that is ready to be mineralised. It is assumed this is the mechanism by which tetracycline, and staining agents like alizarin, are deposited into forming bone [245,246,247], but the movement of such metabolites through these transporters has yet to be demonstrated [259].

A separate, but similar, mechanism for the deposition of polyphenols into bone, could involve the binding of polyphenols to phospholipids in cell membranes prior to their binding to hydroxyapatite calcium. The binding of polyphenols to phospholipids (specifically lipid heads comprising the bilayers of cell membranes) has been proposed as a site of intracellular deposition for circulating polyphenols [263,264,265]. Assuming that polyphenols additionally bind to the lipid bilayer of osteoblasts, there is potential that they are transported to the mineralizing hydroxyapatite surface of bone via matrix vesicles. Given the role of phospholipids in binding and contributing calcium to hydroxyapatite crystals formed inside matrix vesicles [266,267], it may be at this stage that any polyphenols bound to the lipid membrane of vesicles are able to bind to calcium in the forming bone matrix.

It should not be ruled out that some types of polyphenol may also be bound to the protein, predominantly Type I collagen, that is also produced in growing bone. Although there is no strong evidence that this occurs, it is theoretically possible as shown by the binding of polyphenols with proteins such as albumin in circulation [17,268]. The binding of polyphenols with collagen has been studied in biomaterials research, focusing on the ability of polyphenols to form crosslinks within collagen in vitro [269]. Recent work has also investigated the potential proliferative effects that collagen crosslinked with polyphenols may have on bone forming cells or periodontal ligament cells, with further implications for improving bone grafts or tooth replantation [270,271].

The binding affinity of various polyphenols with collagen varies greatly depending on the type of polyphenol [268,272]. While cross-linking can occur with a variety of available compounds, often as sugars [273], other compounds—namely procyanidins and other hydrolysable tannins—exhibit a high affinity for proline, which comprises a substantial part of collagen (~10% by residue for Type 1 collagen) [274]. Polyphenol-collagen binding appears to form stronger cross-links, creating covalent and hydrogen bonding [268,275,276], as well as increasing collagen’s hydrophobicity and protection against collagenases [271,276,277,278]. Gallolyl moieties of hydrolysable tannins are considered the functional groups of polyphenols most capable of forming crosslinks with collagenous proteins, which generally bind with any group capable of forming an H-bond [275,279,280,281]. Gallotannin-collagen crosslinks form bonds with greater hydrophobicity than other tannins due to the orientation of their aromatic rings, C-C bonding, and lack of inter-galloyl linkages, overall providing even greater thermal stability [275,279,280]. However, most studies on the effects of polyphenol-collagen cross-linking have been in vitro, making the potential for this cross-linking in vivo unclear.

Most evidence for absorption of polyphenolic metabolites in soft tissues points to inflammatory stimuli as a trigger [16] and macrophages as specific targets [100], meaning that osteol macrophages may be an alternative pathway into bone. Though osteol macrophages are tied to the process of bone remodeling specifically, and are assumed to be most strongly associated with the removal of apoptotic cells, they remain positioned near bone-lining mature osteoblasts, and may also be associated with regulatory effects in the mineralization phase of bone remodeling [282]. The two proposed mechanisms for the binding of polyphenols to hydroxyapatite, and the bone matrix in general, do not discount the possibility that polyphenols present in extracellular circulation simply bind to remodeling bone without specific interaction with bone remodeling cells [17]. Although the microenvironment of forming bone is tightly regulated with an epithelial barrier formed by osteoblasts [259], the matrix of reabsorbing bone is relatively exposed to the extracellular matrix by osteoclasts, which create an environment with lower pH (≤6.9) [283,284,285], conducive for deconjugation.

### 4.2. The Effect of Polyphenols on Bone Remodeling

Currently there are very little data on the health effects of polyphenols on bone tissue in vivo, and it remains to be discovered whether polyphenolic metabolites found within bone offer anti-oxidant or other beneficial effects during the bone remodeling process. There is, however, interest in the effects of polyphenolic compounds on bone remodeling in diseases that affect bone and cartilage such as osteoporosis [286,287,288,289,290,291]. The pathways and signaling responsible for bone resorption and formation have been extensively reviewed elsewhere (e.g., [290]), the following section provides a brief overview in relation to the effects of polyphenols specifically.

Changes in bone remodeling, with respect to disproportionate increases in bone reabsorption relative to formation, are typically a result of metabolic, age-related, reductions in osteoblast directed formation and, to a more limited extent, promotion of osteoclastic bone resorption [292]. Increased rates of osteoclastogenesis and accumulated damage to lipids and DNA from oxidative stress can accumulate over an animal’s lifetime, leading to increased bone resorption [292]. These age-related processes can be accompanied by significant metabolic changes due to deficiencies in sex hormones that additionally result in greater bone loss and lower overall bone mineral density (BMD) [293,294,295]. As such, metabolic changes to bone cells are exacerbated by oxidative stress with ROS causing a disruption to the remodeling cycle, with free radicals and RNS typically correlating negatively with bone mineral density measurements, and positively with biomarkers for bone resorption markers, particularly among post-menopausal women [132,296,297].

Similar to damage caused in other tissues by oxidative stress, the ROS normally produced in osteoclasts for the purposes of degrading proteins can begin to overwhelm endogenous antioxidant systems (i.e., regulation by GPx) when exacerbated by the aforementioned metabolic changes, leading to increased resorption [298,299,300]. ROS and induced inflammation that causes cellular apoptosis or lipid peroxidation in bone can also increase osteocyte apoptosis, which generally triggers the bone resorption signal [137,138,139,301].

One of the primary natural regulators of bone remodeling are sex hormones, particularly estrogen. Osteoblasts, osteocytes, and osteoclasts all possess receptors for estrogen molecules, as do their precursor cells and most types of cells found within bone tissue [302]. Estrogen appears to play a key role in reducing amounts of interleukin 7 (IL-7) produced from T-cells to inhibit both receptor activator of NF-κB ligand (RANKL) and TNF-α expression, both of which suppress osteoblast differentiation in addition to their promotion of osteoclastogenesis [302,303,304]. Estrogen deficiency then results in the inability to suppress these pathways [302,305]. Estrogen can also promote TGF-β expression in osteocytes as another means of inhibiting RANKL expression [306]. Expression of GSH to regulate ROS is additionally reduced in animals that are deficient in estrogen [300].

Interest in polyphenols for protective or therapeutic applications to bone health stem from their anti-oxidant effects and their seemingly unique ability to regulate cell signaling similar to sex hormones [307]. Polyphenols are seen as a potential replacement for the age-related loss of endogenous anti-oxidant capacity and sex hormones, reducing oxidative stress and lowering the rate of bone resorption, particularly in individuals suffering osteoporosis [288]. How effective polyphenolic compounds are in osteoporosis prevention is debateable, as a survey of currently available literature found that healthy human volunteers given polyphenol doses (often in excess of their traditional dietary regimen), typically experienced no changes in bone metabolism [308]. On the other hand, numerous in vitro studies, and a more limited number of experimental in vivo studies performed on rats, provide promising results for the activation and inhibition of osteoblasts and osteoclasts respectively via polyphenols [289,307,309,310,311]. These data suggest a better understanding of the fate of polyphenols and their ability to target bone cells specifically is prescient.

In vitro study indicates that polyphenols have the potential to reduce bone loss by suppressing pathways resulting in RANKL leading to osteoclast differentiation, either as a result of estrogenic activity, separate forms of gene expression inducing osteoblast differentiation, or by direct scavenging of ROS [289,312,313,314]. Polyphenol capabilities extend to reducing the expression of pro-inflammatory markers, such as COX-2 and TNF-α, which has been experimentally demonstrated on rats given blueberry polyphenols [315]. Reduction of those same markers have been correlated with increased BMD in rats given green tea polyphenols [309]. More directly related to osteogenesis, rats fed polyphenol rich diets of blueberries demonstrated lower RANKL expression from marrow stromal cells and a reduced rate of bone resorption [310]. Polyphenols found in plums, and oleuropein (typically found in olives), have been shown to increase expression of runt-related transcription factor-2 (RUNX2), alkaline phosphatase (ALP), and osterix gene expression in marrow stem cells and bone lineage cell lines, which regulate osteoblast differentiation and bone matrix formation [316,317].

In addition to the majority of studies investigating polyphenols and bone health, which generally find correlations between inflammatory or osteoprogenitor biomarker changes and polyphenol dietary concentrations or changes in vitro, are studies that appear to more clearly show the changes to upstream pathway signalling made by polyphenols. These types of upstream changes have most clearly been demonstrated by the binding of the polyphenol fisetin to MAPK Phosphatase-1 (MKP-1) in order to stabilize it and reduce its degradation in the proteosome [307]. Nuclear factor kappa B, MAPK p38, JNK and further downstream RANKL pathways for osteoclast differentiation were all downregulated in cell lines after stabilizing MKP-1, while oral administration of fisetin appeared to counteract some bone loss in mice that had induced inflammation or had been overactimized [307]. Conversely, Chen and colleagues [318] found that rats fed a blueberry phenolic serum exhibited elevated expression of MAPK p38 (which is implicated in both osteoblast and osteoclast differentiation pathways; [319,320]), leading to the downstream activation of the canonical Wnt/β-catenin pathway for RUNX2 transcription in osteoblasts.

Green tea polyphenols, namely epigallocatechin gallate, have been particularly singled out as stimulating osteoblast differentiation whilst reducing osteoclast differentiation [291]. Epigallocatechin gallate appears to stimulate bone morphogenetic protein-2 (BMP-2) in mesenchymal stem cells to begin osteoblast differentiations and promotes the expression of RUNX2 genes, osteonectin, and osteocalcin, ultimately promoting mineralization [289]. Given that its osteogenic properties do not appear linked to free-radical scavenging, EGCG is likely binding to certain proteins upstream to stimulate this pathway. Byun and colleagues [311] have proposed that green tea polyphenols may induce higher amounts of phosphoroprotein phosphatase, which dephosphorylates transcriptional coactivator with PDZ-binding motif (TAZ), causing it to stabilize and undergo nuclear localization instead of binding to a scaffolding protein and experiencing proteasomal degradation. Stabilized TAZ is subsequently able to regulate transcription factors leading to RUNX2 expression.

The potential for polyphenols to protect against lipid peroxidases specifically offers additional protection to bone cell signaling. The indiscriminate ability of lipid peroxidases to cause the degradation of lipids means that any reduction in high-density lipoproteins (HDL) can disrupt the ability for mesenchymal stem cells to differentiate into osteoblasts [321,322]. Likewise, omega-3 fatty acids are capable of activating the β-catenin signaling pathway (similar to the canonical Wnt pathway) responsible for osteoblast differentiation by binding to the G-protein coupled receptor free fatty acid receptor 4 (GPR120/FFA4), and may additionally inhibit osteoclast activity [323,324].

Further regulation of osteoblast and osteoclast cell growth appears to occur through immunomodulatory regulation via the gut microbiome [241,242]. It has recently been suggested that gut bacteria (in healthy, pathogen-free mice) may suppress the osterix (Sp7) involved in osteoblast differentiation by downregulating insulin-like growth factor (Igf1) in bone, while simultaneously modulating RANKL activity required for osteoclast differentiation [242]. In a general sense, the gut microbiome may have downstream signaling effects that regulate signals involved in bone remodeling [242]. This implies that regardless of bioavailability of polyphenolic compounds beyond the gastrointestinal tract, they may be capable of affecting bone health through localized activity in the gut during digestion, and potentially during metabolism in the liver given the changes in hormones stemming from there.

Many of the in vitro studies on the effects of various polyphenols on osteoblast/osteoclast differentiation have been useful means of identifying possible mechanisms for such activities e.g., [289,311]). Clinical studies have generally not exhibited this level of clarity (often limited to measurements like BMD). Conversely, some experimental rat studies identify in vivo changes in bone remodeling under conditions of induced bone loss which provide information on changes to bone formation/resorption biomarker levels on polyphenol supplemented diets [307]. Considering the evidence for the binding of polyphenol metabolites to bone in vivo [17], further research may be able to link the presence of polyphenol metabolites in bone tissue to other metabolic changes in bone (or other tissues), as well as examine whether these changes are tied to specific polyphenolic compounds. Additional bone-polyphenol studies related to larger animals, such as pigs, may offer insight into how skeletal tissue of different physiologies are affected by the metabolism and presence of polyphenols in diet.

### 4.3. Polyphenols in Bone, Potential for Forensic and Palaeodietary Studies

In palaeodietary studies isotopic changes, preserved in human and animal bone collagen and bioapatite, have been used to interpret dietary information from an individual’s skeletal remains [325,326,327]. However, information from isotopic composition provides information averaged across the diet and over time and does not provide direct evidence for the consumption of specific plants types for example. A small number of archaeological studies have detected compounds similar to polyphenols in the bones of ancient humans, highlighting their presence as biomarkers for the consumption of specific plant-based foods and potentially the therapeutic benefits they may have offered to people in the past. Those studies specifically identified alizarin, likely derived from the consumption of root madder, and tetracycline, likely derived from consuming *Streptomycetes* spp. contaminated grains, in skeletal remains [245,246,247]. These findings indicate polyphenolic compounds (already detectable in modern bone; [17]) may be a source of useful information in modern and archaeological bone as well. Being able to identify diet-derived polyphenolic compounds in bone could be an invaluable means of identifying plant-based dietary sources, beyond the conventional biomolecular approaches currently used (e.g., stable isotope analysis) in palaeodietary and forensic studies [328]. Understanding the role of diet in the potential health changes that coincide with shifts in human and other animal gut microbiome populations (such as changes associated with the industrial revolution) has become of increasing interest in palaeogenomics [329]. Metagenomic research focusing on changes in health could be further refined by incorporating metabolomic analyses targeting polyphenols to identify the foods related to such shifts, particularly within the context of the health effects often associated with polyphenols, both systemically (Section 3) and localized in bone (Section 4.2). In the context of farm animals such as pigs, analyses of polyphenolic compounds in ancient animal bones could improve understanding of shifts in their health related to environmental changes and human practices associated with foddering and animal care.

## 5. Conclusions

Polyphenols are anti-oxidants capable of scavenging ROS, and can reduce the damaging effects of oxidative stress and inflammation in animals. Further research has highlighted how polyphenols are pleiotropic, having additional metabolic effects such as gene regulation and also reducing inflammation, being chemopreventative, and having antibiotic effects. Metabolic processes in vivo can alter and limit the therapeutic benefits of polyphenols. Therefore, being able to link the efficacy of the myriad of possible beneficial and therapeutic health effects of polyphenols demonstrated in vitro to various animal tissues in vivo represents an important area of research.

After ingestion of dietary polyphenols there follows a pattern of Phase I and II metabolism wherein their primarily glycosidic forms, derived from plant sources, undergo hydrolysis and other metabolic changes via enzymatic action in the gastrointestinal tract, tied into the activity of the gut microbiome. The majority of dietary polyphenols and their derivatives are either excreted as feces or undergo conjugation reactions to facilitate their absorption, before further conjugation reactions in the liver produce the metabolites that ultimately end up in circulation and are eventually excreted in urine. The fate of circulating polyphenol metabolites can vary significantly, as do the earlier metabolic processes taking place in the gut, as some dietary polyphenols can be absorbed deconjugated as aglycones and others have been identified in non-intestinal tissues in aglycone form as well. Structural diversity influences absorbance and metabolism of polyphenols, as does inter-individual variations in host metabolism (i.e., affecting what metabolites are produced from dietary polyphenols). Polyphenolic metabolites reach different tissues via circulation and can enter cells as conjugated metabolites that undergo intracellular deconjugation, with mitophagy and inflammation and/or special transport proposed as triggers, although these processes are not yet well understood. The presence of aglycone polyphenol metabolites in bone raises further questions as to the mechanisms responsible for their presence in various tissues.

The connection between beneficial health effects of polyphenols in vitro and health-related effects of dietary polyphenols in animals is unclear, and comparatively understudied in farm animals compared to lab animals and humans. We focused in this review on polyphenol activity and therapeutic responses in pigs, which generally exhibit reductions in inflammation of the gut when challenged by stress and pathogens. Studies have found that pigs experience changes in microbial populations of the gut with the reduction of pathogenic species and increases in prebiotic species when consuming polyphenol supplemented feeds. Examining therapeutic responses to the consumption of polyphenols in pigs has implications for other species, in that changes in biomarkers associated with ROS and inflammation in pigs are shown to be reduced systemically in tissues outside of the gut, even if polyphenol metabolites were not always detected in those tissues (though this does not suggest that they were not present at some point). The potential importance of polyphenols to livestock health lies partly in their ability to act as natural anti-biotics capable of anti-microbial action against pathogenic species. It is not well understood if the anti-microbial activity of polyphenols drives changes in gut microbial species populations, or how related such changes are to the anti-oxidant or anti-inflammatory capabilities of polyphenols acting on the tissues of the gut. It is important to consider how the changes in gut health due to dietary polyphenols leads to systemic therapeutic responses by altering the gut microbiome, and how this affects polyphenol metabolism and absorbance as part of a two-way stream of modulation.

Future work on the health effects of polyphenols and their metabolism requires continued in vivo studies and attempts to refine knowledge of the mechanisms behind the intracellular presence and actions of polyphenols providing anti-inflammatory and antibiotic effects. This includes understanding inter-individual variations in the metabolism of dietary polyphenols and their bioavailability, alongside the stimuli that elicit their health effects and result in the targeting of different tissues and intracellular deconjugation (e.g., the replicability of responses and metabolite/biomarker concentrations in tissues). In vivo polyphenol responses may be the result of testable conditions, such as types of polyphenols consumed, timing of their consumption, the dosage of polyphenols, and their associated food matrix and other sources of nutrition (i.e., combinations of polyphenols or other macromolecules). Approaches that look to multi-omics may also find novel sources of information on the mechanisms behind polyphenol activity and metabolic responses in various cell types under different conditions in vitro that can be examined in vivo.

Bone presents one particular tissue for future research on the metabolic fate of dietary polyphenols, as there is clear evidence of accumulation of at least some types of polyphenols in bone. What is unknown is how or whether polyphenol metabolites are taken up by bone forming or resorbing cells, what different types of polyphenols may bind with bone in vivo, and whether the presence of polyphenols in bone has any effect on, or other relationship with, bone health (including the treatment of bone diseases like osteoporosis). For archaeological and forensic sciences, the detection of polyphenolic compounds in bone has exciting implications for inferring specific sources of plant-based foods in diet and possible forms of cultural or health-related plant-food consumption.

## Figures and Tables

**Figure 1 pathogens-11-00770-f001:**
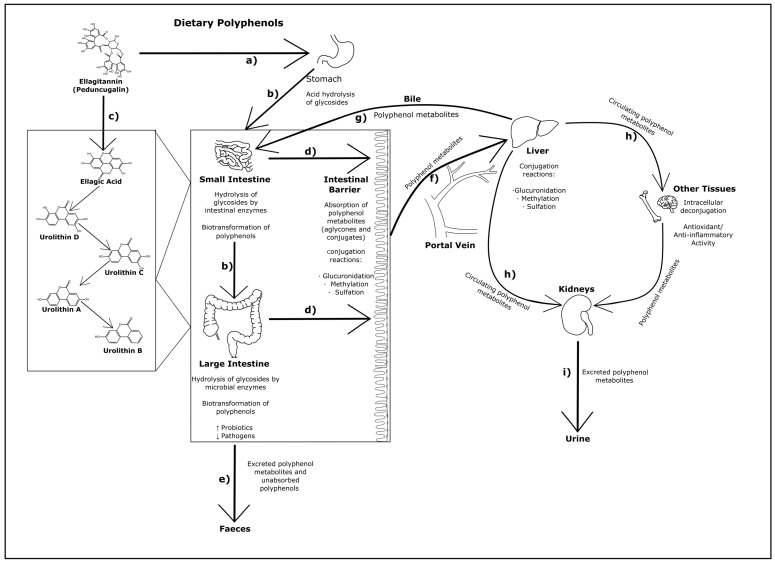
A schematic showing the general biotransformations of dietary polyphenols in monogastric animals: (a) dietary polyphenols (often in the form of glycosides) are initially hydrolyzed by stomach acids; (b) further hydrolysis and biotransformation occurs to polyphenols in the small and large intestines via intestinal and microbial enzymes, effecting changes in the microbial species; (c) The left side of the image depicts the types of structural changes that can occur to polyphenols (e.g., peduncugalin, an ellagitannin) in the gastrointestinal tract; (d) biotransformed dietary polyphenols are absorbed through the intestinal barrier and typically undergo conjugation reactions; (e) remaining polyphenols in the large intestine (both metabolized and unmetabolized) are excreted as feces; (f) absorbed polyphenols are transported via the portal vein into the liver to undergo further conjugation reactions; (g) a portion of polyphenol metabolites re-enter the gastrointestinal tract from the liver via the bile duct; (h) the rest of the polyphenol metabolites in the liver enter circulation, with some reaching cells of body tissues and organs not pictured here (e.g., heart, muscle, brain, bone). Within these tissues there is evidence of polyphenol metabolites deconjugating into aglycones and the activation of anti-oxidant/anti-inflammatory effects; (i) circulating polyphenols are then ultimately excreted in urine via the kidneys (adapted from ([35,36,37]).

**Table 1 pathogens-11-00770-t001:** Classifications and structures of polyphenols (in aglycone form), and their common food sources [12,18,20,21].

Class	Example Compounds	Example Structure	Example Food Sources
**Flavonoids**			
Anthocyanidins	Cyanidin, Delphinidin, Malvidin, Pelargonidin, Peonidin	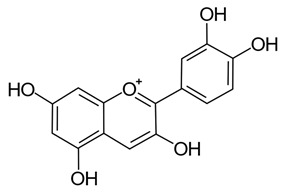 (Cyanidin)	Aubergine, Blackcurrant, Blueberry, Black grape, Cherry, Elderberry, Grape, Orange, Plum, Red wine, Rhubarb, Strawberry
Flavanols	Catechins (Epicatechin, Epigallocatechin, Epigallocatechin gallate, Gallocatechin), Procyanidins	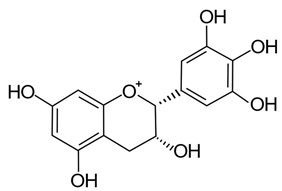 (Epigallocatechin)	Apricot, Black tea, Blackberry, Cherry, Chocolate, Grape, Green tea Legumes, Peach, Teas
Flavones	Apigenin, Luteolin	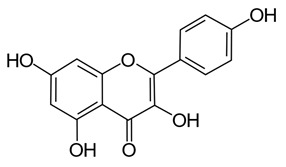 (Apigenin)	Capsicum pepper, Celery, Oregano, Parsley, Rosemary
Flavonols	Kaempferol, Myricetin, Quercetin,	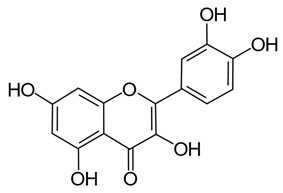 (Quercetin)	Apple, Apricot, Black tea, Broccoli, Green bean, Green tea, Kale, Leek, Onion, Red wine, Tomato
Flavonones	Hesperetin, Hesperidin, Naringenin	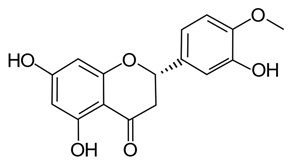 (Hesperetin)	Citrus fruits, Grapefruit, Orange, Peppermint, Tangerine, Tomato
Isoflavones	Genistein, Daidzein, Glycitein	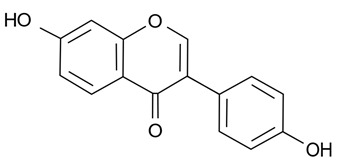 (Daidzein)	Lentils, Soy
**Non-flavonoids**			
Phenolic Acids	Hydroxycinnamic acids (Chlorogenic acid, Caffeic acid, Ferulic acid, Sinapic acid) Hydroxybenzoic acids (Gallic acid, Protocatechuic acid)	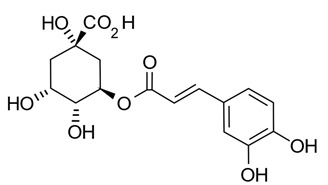 (Chlorogenic Acid)	Artichoke Aubergine, Apple, Blackcurrant, Black tea, Blackberry, Blueberry, Cereals, Cherry, Coffee (Chlorogenic Acid), Kiwi, Pear, Plum, Potato, Raspberry, Red wine, Strawberry
Lignans	Secoisolariciresinol, Matairesinol	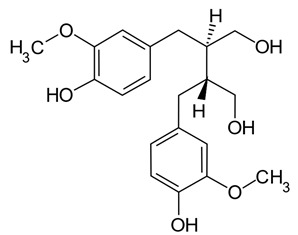 (Secoisolariciresinol)	Linseed, Sesame seed
Hydrolysable Tannins	Ellagitannins (Pedunculagin, Punicalagin) Gallotannins	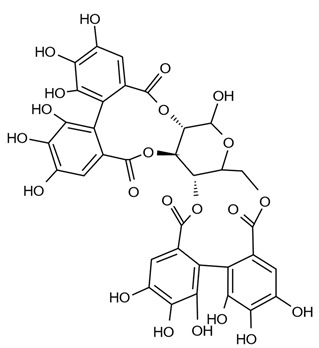 (Pedunculagin)	Walnut, Acorn, Oak, Pomegranate
Stilbenes	Resveratrol	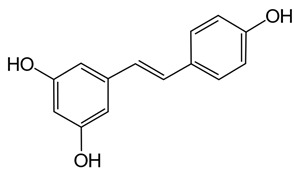 (Resveratrol)	Blueberry, Red grapes, Red wine, Raspberry

## Data Availability

For the purpose of open access, the authors have applied a CC BY public copyright license to any Author Accepted Manuscript version arising from this submission.

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
