# Peer review of "Polyphenols: Bioavailability, Microbiome Interactions and Cellular Effects on Health in Humans and Animals"

_pathogens, 2022, doi:10.3390/pathogens11070770_

Round 1
Reviewer 1 Report
This work presents a comprehensive resource about the bioavailability, effects and metabolic interactions of polyphenols. Writing style is adequate but there are numerous errors in the taxonomic nomenclature of bacteria, and typos. Please revise all the names and their writing convention accordingly.
Author Response
Reviewer: 1 - Comments to the Author
“This work presents a comprehensive resource about the bioavailability, effects and metabolic interactions of polyphenols. Writing style is adequate but there are numerous errors in the taxonomic nomenclature of bacteria, and typos. Please revise all the names and their writing convention accordingly.”
We thank the reviewer for picking up on errors in taxonomic nomenclature and typos in the manuscript. We have carefully proof-read the text and been through each use of taxonomic terminology and applied the correct terminology to the best of our knowledge. Specific alterations are on pages 11-13 (lines 143, 145-149, 153, 157-158, 160, 164, 177, 187-192, 197, and 201), page 27 (lines 520, 522), and pages 30-31 (lines 580 and 591). Alterations are also provided as tracked changes in the updated manuscript.
Reviewer 2 Report
Natural antioxidants role in health related context are highly assayed topic in nowadays. Polyphenols are widely used ingredients in specific dietary ratio in several species. Mostly these compounds have a scavanging function against reactive oxygen species. Polyphenols benefits can be measured in their anioxidant capacity, anti-cancer, anti-microbial and anti-inflammatory activities. Several stress related situation polyphenols have an advanced effect on maintaining homeostasis of especially young animals. Weaning of piglets one of the most stressfull event of their life, in which their digestive systems are highly sensitive against disease related micro-organism overpopulation. From this year withdrawal of ZnO as preventive feed additive at this stressfull period of their life natural replacer of this additive is highly demanded. Polyphenols could be one of the option to use to prevents healthy microbiome in this life phase of them. Further use and role pf ponyphenols were highlighted in the article in bone remodelling which is has high impact in human too.
This manuscript is a very well organized, easy readable and welll understanding review about the function of polyphenols benefical and potential use in healthy diet in animals and human too.
Author Response
Reviewer: 2 - Comments to the Author
“This manuscript is a very well organized, easy readable and welll understanding review about the function of polyphenols benefical and potential use in healthy diet in animals and human too.”
We thank the reviewer for their positive feedback on our manuscript.
Reviewer 3 Report
This review paper is focused on chemically modification of polyphenols both by the gut microbiome and, once absorbed into circulation, how this may be linked to subsequent therapeutic effects. In the next part, the work deals with the transport of polyphenols to different tissues and the mechanisms for their intracellular uptake. The strength of the study lies in the detailed classification of polyphenols and also in the well-described metabolism of dietary polyphenols. However, the work lacks an explanation of epigenetic mechanisms, and it is known that polyphenols affect these processes. Although the authors state on page 12 that polypenols may alter gene regulation (lines 408-411), this is not a sufficient explanation. This is also the weakness of the study.
Comments:
I recommend supplementing the influence of polyphenols on epigenetic regulation and how it can be used in the treatment of cancer.
Author Response
Reviewer: 3 - Comments to the Author
The strength of the study lies in the detailed classification of polyphenols and also in the well-described metabolism of dietary polyphenols.
We thank the reviewer for their positive feedback on the strengths of our manuscript.
"However, the work lacks an explanation of epigenetic mechanisms, and it is known that polyphenols affect these processes. Although the authors state on page 12 that polypenols may alter gene regulation (lines 408-411), this is not a sufficient explanation. This is also the weakness of the study. I recommend supplementing the influence of polyphenols on epigenetic regulation and how it can be used in the treatment of cancer."
We thank the reviewer for the suggestion to add some explanation of epigenetic effects and mechanisms in relation to polyphenols. We agree with this suggestion and have now reconfigured and supplemented the section on anti-oxidant effects of polyphenols (Section 3.2) to include additional focus on the discussion of epigenetic effects. This can be found on pages 22-25, and is reproduced below for convenience:
“3.2 The anti-oxidant activity and epigenetic effects of plant-derived polyphenols
The antioxidant activity of polyphenols in vivo is thought to be mainly via the scavenging ROS, in a similar way to endogenous anti-oxidants. It has been shown that polyphenols can inhibit enzymatic activity that causes the production of ROS both in vitro and in vivo, although the mechanisms are not fully understood [121,142–145]. The chelating properties of polyphenols are also of notable importance in their ability to relieve oxidative damage brought on by circulating metals in incidences of oxidative stress. Polyphenols with catechol groups are often very good metal chelators as they are able to form bidentate ligands, and similarly, polyphenols with hydroxyl groups in peri positions on phenolic rings also make strong candidates for chelation, although pH is a key factor in their ability to bind with metals [146,147]. Fenton reactions with hydrogen peroxide and iron or copper ions can create superoxides capable of damaging biomolecules like DNA [148]. Polyphenols with catechol and/or galloyl groups that scavenge free radicals form semi-quinone compounds that are further capable of reducing Fe3+ ions, causing the polyphenols to become pro-oxidant quinones [148]. The ability for polyphenols to chelate may be hampered if they are conjugated with other moieties, such as sugars [146], and conversely, the anti-oxidant capabilities of polyphenols appear to be reduced in the presence of metals, as metal ligands may reduce their reactivity with free radicals [147,149]. Most of the evidence for the reduction of ROS via chelation of polyphenols with iron are based on in vitro studies, and a fairly robust body of evidence suggests that dietary polyphenols may ultimately inhibit the absorption of dietary iron [150]. Not enough evidence is currently available to indicate whether dietary polyphenols have a significant effect on the absorption of calcium in the gut, which may be of relevance to the mobilisation of calcium to extent tissues, such as bone (Section 4.1).
Beyond their capability for scavenging ROS, polyphenols can also affect gene regulation and activate transcription factors responsible for triggering endogenous anti-oxidants, such as nuclear factor erythroid 2-related factor 2 (NRF2) [151,152]. Altered gene regulation by polyphenolic compounds appears to extend to the suppression of transcription factors responsible for inflammation and tumour formation, such as activator protein 1 (AP-1) [153], as well as direct inhibition of proinflammatory cyclooxygenases (COX) and ROS lipid peroxidases [154]. Anti-oxidant and anti-inflammatory effects of polyphenols are often observed as an increase in the anti-oxidant capacity of tissues, typically in the form of increased concentrations of GPx and/or SOD, reductions in pro-inflammatory cyclooxygenases and malondialdehydes (a product of lipid peroxidation from ROS) [64].
Some polyphenols have direct chemo-preventative effects that go beyond their antioxidative effects [10,155–157]. For example, ellagitannins and ellagic acids (including their metabolized derivatives) have been shown to promote the activation of apoptotic pathways, such as increasing mitochondrial caspases (e.g., cyto C and Caspase 9) in human cancer cell lines and reducing their expression in non-cancer cell lines [106,141]. Ellagitannin derived metabolites may alternatively regulate the necessary cyclins (downregulating cyclins A and B1, upregulating cyclin E) for S-phase arrest to cause apoptosis in cancerous cells as well [106,141]. The anti-proliferative effects of ellagitannins towards tumour cells may be in part due to the activation of tannase-related genes [61]. Overall, a range of polyphenolic compounds appear to be capable of regulating some of the major pathways involved in tumour formation, including the p53 tumour suppression gene and through inhibition of MAPK pathways that can lead to cancer cell growth [123]. Reductions in biomarkers for inflammation and tumor formation likely extend from the epigenetic effects of polyphenols and their modulation of microRNA (miRNA) expression, DNA methylation, and histone acetylation and/or methylation [158–160].
Inhibition of DNA methyltransferase in human cancer cell lines when supplemented with dietary polyphenols has been shown to result in the demethylation of promoters for tumor formation or the reactivation of tumor suppression genes [161]. In these instances, polyphenols, such as epigallocatechin gallate, can inhibit DNA methyltransferase 1 (DNMT1) by binding with protein residues in a cytosine active site, preventing the entry of DNMT1 [162]. Expression of miRNAs responsible for inactivation of tumor suppression genes in cancer cell lines are also reduced in the presence of polyphenols such as resveratrol [163] and oleuropein [164]. Oleuropein, and the products it is derived from, namely olive oil, have been studied extensively for their epigenetic effects [165], and alterations in miRNA expression that coincide with increased DNA methylation are mitigated and reversed in in vivo rat models supplemented with olive oil [166]. Proliferation of human colon adenocarcinoma cells (Caco-2) supplied with separate treatments of extra virgin olive oil, an olive oil phenolic extract, and hydroxytyrosol, was reduced alongside increases in type-1 cannabinoid receptor (CB1) as a result of inhibition of DNA methylation at the cannabinoid receptor 1 (CNR1) promoter [167]. A similar decrease in CNR1 promoter methylation and an increase in CNR1 expression was observed in rats administered extra virgin olive oil via gavage [167].
Additional mechanisms for polyphenol immunological and anti-cancer effects appear to result from binding with cell receptors or the inhibition of histone acetylation. For example, epigallocatechin gallate can bind with cell receptors, such as zeta-chain-associated 70kDA protein (ZAP-70) and the 67kDA laminin receptor (67LR), the latter of which is expressed in cells involved in immune responses, such as monocytes/macrophages, mast cells, and T-cells [168,169]. Binding of epigallocatechin gallate to 67LR may inhibit human colon adenocarcinoma cell growth [170], or result in apoptosis of multiple myeloma cells [168,171]. The proposed mechanism for these anti-cancer effects is through binding of epigallocatechin gallate to 67LR, and the inhibition of myosin II regulatory light chain (MRLC) or extracellular-signal regulated kinase (ERK) 1/2 phosphorylation and cytokinesis [170,172], which similarly leads to a reduction of histamine release as a response to allergy diseases [172]. Quercetin can inhibit p300 histone acetyl transferase (HAT) activity, decreasing acetylation of nuclear factor kappa B (NF-κB) and levels of inflammatory and tumor promoting enzymes (COX-2, TNF) [173,174]. Suppression of p300 HAT activity via quercetin has also been observed as decreasing acetylation of histone H3 promoter regions of the interferon gamma inducible protein 10 (IP-10) and macrophage inflammatory protein 2 (MIP-2), mitigating inflammation in intestinal epithelial cell lines [175]. In mice with an induced colorectal cancer cell model, in vivo reductions of biomarkers for tumor formation and colorectal cancer cell proliferation have also been observed following administration of polyphenols derived from foxtail millet, alongside renewal of gut microbiome diversity to that of normal mice [176].
The anti-oxidant capabilities of polyphenols have a knock-on effect of regulating micro-environments by scavenging ROS and reduce oxidative stress, resulting in reductions in inflammation, or cancer cell proliferation and the effects of other metabolic diseases (Section 4.2), while further being capable of directly interacting with proteins, enzymes, and other metabolites within immunological and metabolic pathways. The relationships between epigenetic effects of polyphenols and immune responses, anti-cancer effects, and other diseases have been reviewed extensively [158,159,169,177]). Despite this, the effects of various polyphenols in vivo are not as well tested outside of murine models, although there has been an increased focus on how supplementation of farm animals such as pigs with dietary polyphenols can result in increases in anti-oxidant activity, reductions in inflammatory biomarkers, and changes in microbiome species populations (Section 3.4).”